# Development Trend and Driving Factors of Agricultural Chemical Fertilizer Efficiency in China

**Rong He [1], Chaofeng Shao [2,*] , Rongguang Shi [3], Zheyu Zhang [2] and Run Zhao [2]**

[1]   Sichuan Academy of Environmental Policy and Planning, Chengdu 610041, China; heenhr@163.com
[2]   College of Environmental Science and Engineering, Nankai University, Tianjin 300071, China;
      2120170639@mail.nankai.edu.cn (Z.Z.); 2120190607@mail.nankai.edu.cn (R.Z.)
[3]   Agro-Environmental Protection Institute, Ministry of Agriculture and Rural Affairs, Tianjin 300071, China;
      winsomesky@163.com
*   Correspondence: shaocf@nankai.edu.cn; Tel.: +86-1382-068-5571

**Abstract:** In China, a high input of chemical fertilizers is currently required for a relatively low increase in agricultural production, and this has resulted in prominent nonpoint source pollution and problems related to the quality of agricultural products. These phenomena threaten China's implementation of United Nations Sustainable Development Goal 2 (SDG-2). To explore agricultural chemical fertilizer efficiency and the factors driving the growth in chemical fertilizer use in China, as based on an international comparative analysis of China's chemical fertilizer input, the development trend in the application level and the efficiency in the use of chemical fertilizer in China were subject to time series analysis, and the factors influencing change were identified and analyzed using the Logarithmic Mean Divisia Index (LMDI). The results show that: (1) The gap in agricultural chemical fertilizer efficiency is still large when comparing China with modern agricultural countries, and excessive fertilizer input is still a major problem. (2) The continuous growth in the total amount of chemical fertilizer applied in China during the past 18 years has contributed to the increase in chemical fertilizer application intensity, which provided a cumulative contribution of 85.52%, with smaller contributions from the planting structure and crop sown area. Based on the analysis of fertilizer application, the chemical fertilizer application intensity of the main grain crops was the most significant factor, accounting for about a 40.00% cumulative contribution. (3) Since 2015, the total amount of chemical fertilizers has been reduced through gradually improving the utilization rate of chemical fertilizers, reducing the application intensity of chemical fertilizers, and implementing the fallow rotation system and other measures. Of these, the reduction in application intensity was the most effective at reducing the overall amount of applied fertilizer. To meet the target for achieving sustainable agricultural development, China must still reduce its use of chemical fertilizers by at least 21.80 million tons. Based on the results of current measures and international experience, some suggestions for reducing fertilizer usage are provided.

**Keywords:** use efficiency; LMDI; decoupling; reduction target; SDG-2

## 1. Introduction

Fertilizer application contributes to more than 40% of grain production [1,2], which is important for China. The grain production uses 9% of the world's arable land, produces 24% of the world's grain, and feeds 18% of the world's population [3]. Chemical fertilizers have considerably contributed to alleviating the discordance between China's limited arable land resources and the huge demand for food, but excessive application of chemical fertilizers has led to the serious nonpoint source pollution of farmland and problems with the quality and safety of agricultural products. Chemical fertilizers have become the main source of nitrogen and phosphorus in China's surface water, the main cause of nitrate

pollution in groundwater [4], and the main contributor to heavy metal pollution in soil [5], threatening China's implementation of United Nations' Sustainable Development Goal 2 (SDG-2: to end hunger, achieve food security and improved nutrition, and promote sustainable agriculture). To reduce the environmental pollution caused by the excessive use of chemical fertilizers, developed regions, such as the European Union [6], the United States [7–10], and Japan [11] started to reduce the use of chemical fertilizers and initiated research on and treatment of agricultural nonpoint source pollution in the late 1980s, which achieved remarkable results. However, the path used to reduce chemical fertilizers in developed countries is unsuitable for developing countries due to the huge differences in population density, agricultural production demand, economy, resources, and other national conditions. In 2015, China established an action plan for achieving zero growth in chemical fertilizer use for major crops by 2020, providing new models and samples for solving the chemical fertilizer environmental pollution globally and sharing their experience with developing countries. The direction and countermeasures to reduce the amount of applied chemical fertilizers must be accurately defined to provide the basis for realizing the goal of zero growth in fertilizer application by 2020, by analyzing the development trend in Chinese fertilizer application and its influencing factors, elucidating the influencing mechanism of various factors and calculating the reduction target of Chinese fertilizer use.

Scholars have actively studied the factors driving the changes in the amount of fertilizer application. Wang et al. [12] used the index decomposition method to analyze the reasons for the increase in agricultural fertilizer application in China. Based on the planting area and fertilizer consumption per unit area data, Xin et al. [13] found that the main reason for the strong growth in fertilizer consumption in China is the increased fertilizer use for horticultural crops. Zhang et al. [14] studied the relationship between fertilizer consumption and planting structure. Huang et al. [15] explored the reason for the excessive application of fertilizers from the perspective of farmer behavior. Using factor decomposition analysis, Luan Jiang et al. [16] found that the intensity of fertilizer application is the reason for the continuous increase in agricultural fertilizer application in China. However, most of these studies focused on the correlation of one or several factors with fertilizer application, which cannot fully reflect the contribution of each individual factor to the increase in agricultural fertilizer application. On the basis of international comparative analysis of fertilizer application and time series analysis of fertilizer output rate, we used the logarithmic mean Divisia index (LMDI) to explore the main factors affecting the total increase in fertilizer application since the 21st century and to verify the progress and effects of implementing the zero increase in fertilizer use by 2020 action plan. The findings provide a scientific basis and policy suggestions for reducing fertilizer application in China and for these programs, as developed in China, to be adopted more widely for the sustainable development of world agriculture.

## 2. Research Method

### 2.1. Calculation Method of Chemical Fertilizer Use Efficiency

Scientific evaluation of the effect of fertilizer application is important for improving the fertilization technology, increasing the efficiency of fertilizer resource use, enhancing agricultural production and efficiency, and ensuring the sustainable development of agriculture. The main methods and indexes used to evaluate the chemical fertilizer use efficiency are the apparent recovery efficiency, agronomic efficiency, and partial factor productivity (PFP) of fertilizer application [17–20]. PFP is the ratio of crop yield and fertilizer application amount under the application of a specific fertilizer, which can be used to effectively measure the use efficiency of chemical fertilizers without the need to measure the amount of nutrient absorption. PFP is simple, easy to understand, and has been widely used in the international agricultural community. PFP is calculated as follows:

$$\text{PFP} = \frac{E}{Q} \tag{1}$$

where $E$ is the crop yield and $Q$ is the total chemical fertilizer application.

## 2.2. Decomposition Analysis of Factors Driving Chemical Fertilizer Application Growth

The LMDI method was used to determine the four factors driving the increase in chemical fertilizer application in China from 2000 to 2017: Chemical fertilizer application intensity, planting structure, total crop area, and marginal contribution. The contribution rate of each factor to the application rate of chemical fertilizers and the reasons for the increase in chemical fertilizer application quantity were analyzed, and we sought out countermeasures to reduce chemical fertilizer application.

The decomposition analysis equation for the driving factors of chemical fertilizer application growth is

$$Q_j = \sum_i (I_{i,j} \times A_{i,j}) \tag{2}$$

where $Q_j$ is the total amount of chemical fertilizer application in the $j$th year, $I_{i,j}$ is the intensity of chemical fertilizer application in year $j$ of the $i$th crop, and $A_{i,j}$ is the planting area of the $i$th crop in year $j$.

Considering the effect of planting structure on the amount of chemical fertilizer application, Equation (2) is decomposed into

$$Q_j = \sum_i (I_{i,j} \times P_{i,j} \times a_j) \tag{3}$$

where $P_{i,j}$ is the proportion of the sown area of the $i$th crop in the $j$th year to the total sown area, and $a_j$ is the total sown area of the $j$th crop.

The change in chemical fertilizer application from the base year to the first year of the $i$th crop can be expressed as

$$\begin{aligned} \Delta Q_{i,1} = Q_{i,1} - Q_{i,0} &= I_{i,1} \times P_{i,1} \times a_1 - Q_{i,0} \\ &= I_{i,0} + \Delta I_{i,1} \times (P_{i,0} + \Delta P_{i,1}) \times (a_0 + \Delta a_1) - Q_{i,0} \end{aligned} \tag{4}$$

where $\Delta Q_{i,1}$, $\Delta I_{i,1}$, and $\Delta P_{i,1}$ are the changes in the amount of chemical fertilizer application, the chemical fertilizer application intensity, and the proportion of the sown area of the $i$th crop, respectively; $\Delta a_1$ is the change in the total area of sown crops.

As $Q_{i,0} = I_{i,0} \times P_{i,0} \times a_0$, Equation (4) can be further decomposed into

$$\begin{aligned} \Delta Q_{i,1} &= \Delta I_{i,1} \times P_{i,0} \times a_0 + I_{i,0} \times \Delta P_{i,1} \times a_0 + I_{i,1} \times P_{i,1} \times \Delta a_1 + \Delta I_{i,1} \times \Delta P_{i,1} \times a_0 \\ &= G_{i,1} + L_{i,1} + H_{i,1} + V_{i,1} \end{aligned} \tag{5}$$

where $\Delta I_{i,1} \times P_{i,0} \times a_0$ is the amount of change in chemical fertilizer application caused by a separate change in the intensity of chemical fertilizer application, expressed as $G_{i,1}$; $I_{i,0} \times \Delta P_{i,1} \times a_0$ is the change in the amount of chemical fertilizer application caused by individual changes in the planting structure, which is expressed by $L_{i,1}$; $I_{i,1} \times P_{i,1} \times \Delta a_1$ is the change in chemical fertilizer application caused by changes in the total area sown by the crop and is expressed by $H_{i,1}$; and $\Delta I_{i,1} \times \Delta P_{i,1} \times a_0$ is the change in chemical fertilizer application caused by the joint changes in chemical fertilizer use intensity and planting structure (hereinafter referred to as marginal contribution), which is expressed by $V_{i,1}$.

$$\Delta Q_{i,1} = G_{i,1} + L_{i,1} + H_{i,1} + V_{i,1} \tag{6}$$

$$\Delta Q_1 = \sum_i \Delta Q_{i,1} = \sum_i (G_{i,1} + L_{i,1} + H_{i,1} + V_{i,1}) \tag{7}$$

The proportion of the change in the amount of chemical fertilizer application to the total amount of chemical fertilizer application caused by the change in each driving factor is its contribution to the total change in chemical fertilizer application. The influence of the marginal contribution on the change in total chemical fertilizer application is minimal and, thus, was not considered in this study.

## 2.3. Data Source and Processing

Sustainable agriculture is not simply the direction of agricultural development in the 21st century but also the only way forward for China's agricultural development. In China's implementation of

SDG2 toward the realization of sustainable agricultural development, the overall development trend of chemical fertilizer application level and use efficiency in China must be studied, along with the factors driving the increase in chemical fertilizer application since the beginning of the 21st century. Therefore, given the availability of data, we conducted comparative studies on the intensity of chemical fertilizer application and main grain yield of typical developed countries in 2002–2016, and other studies in 2000–2017.

The data for this study were mainly derived from the International Fertilizer Association (IFA) database [21], the World Bank database (TWB) [22], the Food and Agriculture Organization (FAO) of the United Nations database [23], the National Compilation of Agricultural Product Costs and Benefits (APCB, 2001–2018) [24], the Ministry of Agriculture and Rural Affairs of People's Republic of China (MARA) [25], and China Statistical Yearbook (CSY, 2001–2018) [26] (Table 1).

**Table 1.** Sources of different data types.

| Data Source | Data Type |
|:-----------:|:---------:|
| IFA | Chemical fertilizer application in typical developed countries |
| TWB | Chemical fertilizer application intensity in typical developed countries |
| FAO | Main grain yield in typical developed countries |
| APCB | Yield and chemical fertilizer application intensity of the major crops |
| MARA | Target area of rotation and fallow |
| CSY | Other data |

The fertilizer application data available for analysis are the latest research data (from 2017), and the sown area data are from when before the fallow rotation system began to be used in China (from 2015).

Due to the different sources and definitions of the data, the yield data per unit area of the three grain crops in the National Compilation of Agricultural Product Cost–Benefit and those calculated by the China Statistical Yearbook are not the same, being about 1.10–1.18 times greater than the latter. To reduce the errors caused by data inconsistency, the data were normalized.

## 3. Results and Discussion

### 3.1. Total Chemical Fertilizer Application Level and Use Efficiency in China

#### 3.1.1. International Comparison of the Current Status of Chemical Fertilizer Application

Table 2 shows the development trends in agricultural fertilizer application in the major developed countries around the world. China's total fertilizer use had soared from 34.75 million tons in 2000 to 46.48 million tons in 2017, representing an increase of 33.78% and an average annual growth rate of 1.73%, higher than the rate of 1.24% for the United States. China's total chemical fertilizer application was significantly higher than those of other countries, accounting for about one-quarter of the world's total chemical fertilizer application and twice that of the United States. Different from the overall trend in fertilizer application in China, the total chemical fertilizer application in developed countries, such as Japan, the Netherlands, the United Kingdom, and France, showed negative growth in varying degrees, with average annual growth rates of −1.88%, −2.31%, −1.19%, and −1.67% from 2000 to 2017, respectively.

Figure 1 depicts the intensity of fertilizer application between China and typical developed countries in the world from 2002 to 2016. China's chemical fertilizer application intensity was significantly higher than those of other countries and showed a steady upward trend for 12 consecutive years. After the original Ministry of Agriculture (renamed the Ministry of Agriculture and Rural Affairs in 2018) proposed the action plan of zero growth in fertilizer and pesticide use by 2020, the technology for reducing fertilizer application and increasing its efficiency has been vigorously developed, and the initial results have indicated progress, with the first reduction in fertilizer application intensity observed in 2015. In the same period, the intensity of fertilizer application in the United States remained between

110.00 and 140.00 kg/ha, a relatively low level, and the growth rate and average annual growth rate were also low. Japan, the Netherlands, the United Kingdom, France, and other highly intensive agricultural countries have taken measures to control and reduce the total amount of applied agricultural fertilizer since the 1980s, and the intensity of fertilizer application has declined to varying degrees, with the decline respectively reaching 27.39%, 32.62%, 18.16%, and 22.79% in these countries. In 2016, the application intensity of chemical fertilizers in China was 503.32 kg/ha, which was 3.63, 2.08, 1.74, 1.99, and 3.09 times that of Japan, the Netherlands, the United Kingdom, France, and the United States, respectively, and far higher than 138.9 kg/ha, the average level of chemical fertilizer use in the world.

**Table 2.** Chemical fertilizer application in China and typical developed countries from 2000 to 2017 ($10^4$ t).

| Year | China | USA | Japan | Netherlands | UK | France |
|------|-------|-----|-------|-------------|-----|--------|
| 2000 | 3474.80 | 1879.79 | 145.23 | 41.80 | 184.00 | 414.52 |
| 2001 | 3555.60 | 1961.42 | 135.40 | 41.60 | 187.00 | 416.53 |
| 2002 | 4025.40 | 1935.31 | 130.10 | 40.60 | 178.80 | 396.71 |
| 2003 | 3956.80 | 2120.31 | 131.07 | 39.40 | 178.50 | 398.30 |
| 2004 | 4307.70 | 2009.07 | 129.22 | 35.90 | 168.52 | 390.78 |
| 2005 | 4377.82 | 1927.33 | 129.42 | 34.80 | 159.85 | 353.86 |
| 2006 | 4693.91 | 2077.09 | 129.44 | 35.00 | 155.00 | 349.26 |
| 2007 | 4522.95 | 1945.51 | 123.55 | 35.70 | 157.60 | 382.82 |
| 2008 | 4179.90 | 1604.57 | 99.00 | 31.60 | 125.72 | 278.52 |
| 2009 | 4586.63 | 1888.00 | 95.63 | 31.10 | 145.40 | 289.07 |
| 2010 | 4654.42 | 1979.23 | 110.02 | 30.30 | 147.65 | 343.54 |
| 2011 | 5002.33 | 2036.28 | 104.16 | 25.50 | 145.07 | 293.33 |
| 2012 | 5180.14 | 2086.13 | 104.94 | 25.70 | 145.65 | 314.50 |
| 2013 | 5239.17 | 2137.00 | 103.78 | 26.64 | 154.30 | 308.20 |
| 2014 | 5016.33 | 2039.20 | 109.91 | 24.90 | 151.90 | 305.97 |
| 2015 | 5329.67 | 2125.26 | 101.43 | 27.10 | 149.20 | 301.08 |
| 2016 | 4877.64 | 2254.08 | 103.00 | 28.66 | 151.10 | 298.99 |
| 2017 | 4648.47 | 2317.76 | 105.10 | 28.09 | 150.10 | 311.20 |
| Average annual growth rates | 1.73% | 1.24% | −1.88% | −2.31% | −1.19% | −1.67% |

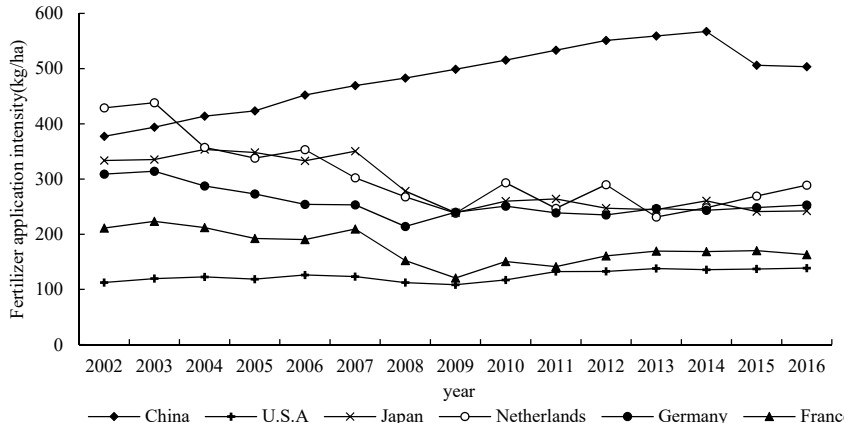

**Figure 1.** Comparison of fertilizer application intensity in typical developed countries from 2002 to 2016.

The increase in fertilizer application intensity can promote the increase in main grain yield, but the promotion effect in China was relatively poor. From 2002 to 2016, China's grain yield increased 21.71% with a 33.35% increase in fertilizer application intensity, while the United States' grain yield increased 52.86% with a 23.18% increase in fertilizer application intensity. In Japan, the Netherlands, Germany, France, and other countries where the intensity of fertilizer application was decreasing, the main grain

yield was generally high, despite fluctuations. In contrast to the level of fertilizer application intensity, the main grain yield in China was lower than that in other typical developed countries, and the average grain yield in the 15-year period was 5627.08 kg/ha, only 0.60–0.90 times that in the United States, Japan, the Netherlands, Germany, France, and other developed countries (Figure 2).

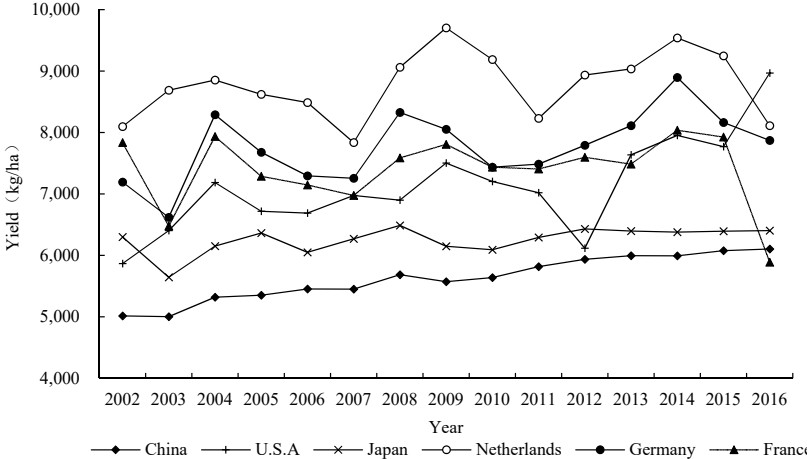

**Figure 2.** Comparison of main grain yield in typical developed countries from 2002 to 2016.

### 3.1.2. Chemical Fertilizer Use Efficiency in China

Figure 3 shows the development trend in grain output and the amount of chemical fertilizer application in China. In general, given the rapid development of the agricultural economy, China's grain output increased continuously in 18 years, with an average annual growth rate of 2.05%. However, compared with the growth rate of total chemical fertilizer application in the same period (about 2.13%), the growth rate was still relatively low. In terms of the change over the past 40 years, China's grain output doubled with the quadrupled increase in the amount of applied chemical fertilizers, which is typical for a country with high input and low efficiency. The development trend of China's grain output and fertilizer application is characterized by three main stages:

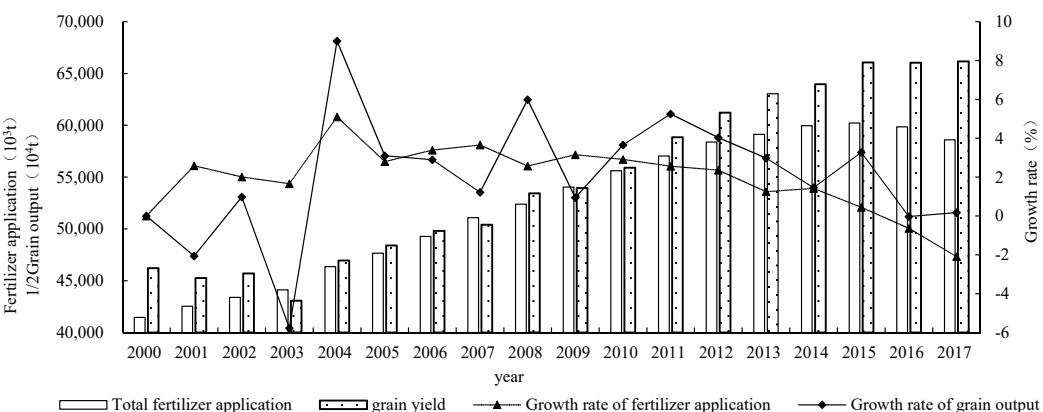

**Figure 3.** Development trend of grain output and fertilizer application in China from 2000 to 2017.

From 2000 to 2003, the grain output declined annually and the per unit yield of grain remained unchanged due to prominent problems with agriculture, rural areas, and farmers; the decline of agricultural income; and the sharp decrease in grain enthusiasm [27,28]. However, the amount of chemical fertilizer applied to the main grain crops increased continuously in the same period, and the application intensity of chemical fertilizer continued to rise.

From 2003 to 2015, China started to implement policies to encourage farmers to grow grain, revitalize agriculture, abolish agricultural taxes, and improve the price of grain purchase and the

level of subsidies [26–28], which led to 12 consecutive increases in grain production, with an average growth of 3.15%. This was, however, accompanied by the rapid growth in the amount and intensity of chemical fertilizer application.

Although chemical fertilizers provide the necessary nutrient elements or protection for crop growth, once the input is close to or exceeds the maximum capacity of the soil and the maximum yield demand of the crop, excess nutrients and pollutants accumulate in the soil. This causes an overload in the operation of the land ecosystem as well as changes in the physical, chemical, and biological characteristics of the soil. These changes damage the ecological environment, and the resulting soil pollution further affects the increase in grain production. Agricultural nonpoint source pollution, soil degradation, and other phenomena caused by the excessive use of chemical products, such as chemical fertilizers, are on the rise, resulting in increasing constraints on China's agricultural resources and environment [29–31]. China has also gradually noted the double-edged sword of chemical fertilizer input in ensuring food security and improving food production capacity, and has attached great importance to the negative impacts of excessive chemical fertilizer application on the ecological environment. After 2015, the original Ministry of Agriculture established an action plan toward reaching zero growth in chemical fertilizer use by 2020 based on the government working policy of stabilizing grain, increasing income, adjusting structure, improving quality, and increasing efficiency to reduce the amount of chemical fertilizer application, and realize the growth of chemical fertilizer application controlled by decoupling, achieving promising preliminary results. In 2016, the amount of chemical fertilizer application decreased for the first time. In 2017, it continued to decline to 58.59 million tons while the grain output remained stable at over 660.00 million tons.

Figure 4 demonstrates the trends in PFP of the main crops in China. In general, China's implementation of measures, such as reducing fertilizer application and increasing its efficiency, soil testing, and formula fertilization, has promoted the overall fluctuation and upward trend of PFP of China's major crops. In 2017, the PFP of the major crops was 18.70 kg/kg, 1.84 kg/kg higher than that in 2000. In terms of crop types, the PFP for rice, wheat, and corn increased from 20.15, 13.17, and 17.10 kg/kg in 2000 to 21.21, 15.31, and 20.16 kg/kg in 2017, respectively, with average annual growth rates of 0.30%, 0.89%, and 0.97%, respectively. Among the three major crops, the PFP of rice was the highest, with an average of 21.40 kg/kg. The PFP of corn was slightly lower than that of rice, with an average of 20.31 kg/kg. The PFP of wheat was the lowest, with an average of 15.15 kg/kg of fertilizer. As early as the end of the 20th century, the PFP of rice and wheat in Japan, the United States, and other developed countries reached more than 24.00 and 18.00 kg/kg, respectively, and the PFP of corn remained around 30.00 kg/kg. A large gap exists between China's PFP and that of developed countries [32].

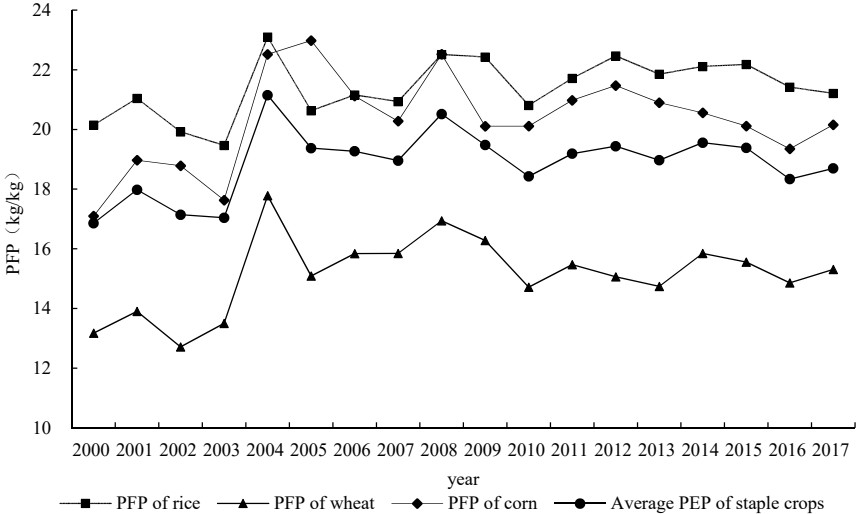

**Figure 4.** Development trends of PFP of main grain crops in China from 2000 to 2017.

### 3.2. Driving Factors for the Increase in Chemical Fertilizer Application in China

#### 3.2.1. Driving Factors for the Growth of Total Chemical Fertilizer Application

The LMDI method was used to analyze the factors driving the growth of and the factors influencing the change in the application of chemical fertilizer in China from 2000 to 2017. The values for the contribution of various factors to the application of chemical fertilizer are listed in Table 3. The cumulative contribution trend is shown in Figure 5. In the examined 18-year period, the amount of fertilizer application in China increased by 17.36 million tons, and the increasing intensity of chemical fertilizer application was the leading reason for the increase in chemical fertilizer application in China, with a cumulative contribution rate of 85.52%. This result highlights the decisive effect of chemical fertilizer application intensity on the increase in total chemical fertilizer application, which was followed by an increase in planting area, with a contribution rate of 21.67%. The cumulative contribution of the adjustment of planting structure to the increase in chemical fertilizer application was 1.25 million tons, with an average contribution rate of −7.19%, which indicates that although the adjustment of the planting structure had a small impact on the application of chemical fertilizer, it benefitted the reduction in chemical fertilizer application.

**Table 3.** Analysis of the factors driving the growth in fertilizer application from 2000 to 2017.

| Time Slot | Total Variation of Fertilizer Application (10⁴ t) | Application Intensity | | Planting Structure | | Sown Area | |
|---|---|---|---|---|---|---|---|
| | | Contribution Value (10⁴ t) | Contribution Rate (%) | Contribution Value (10⁴ t) | Contribution Rate (%) | Contribution Value (10⁴ t) | Contribution Rate (%) |
| 2000–2002 | 210.31 | 324.79 | 154.44 | −66.28 | −31.52 | −48.20 | −22.92 |
| 2003–2005 | 423.42 | 412.95 | 97.53 | −18.93 | −4.47 | 29.40 | 6.94 |
| 2006–2008 | 481.70 | 524.02 | 108.79 | −5.55 | −1.15 | −36.77 | −7.63 |
| 2009–2011 | 461.37 | 263.55 | 57.12 | −17.67 | −3.83 | 215.49 | 46.71 |
| 2012–2014 | 290.03 | 125.32 | 43.21 | −9.59 | −3.31 | 174.30 | 60.10 |
| 2015–2017 | −131.19 | −166.37 | 126.81 | −6.79 | 5.18 | 41.97 | −31.99 |
| Cumulative total | 1735.64 | 1484.26 | 85.52 | −124.81 | −7.19 | 376.19 | 21.67 |

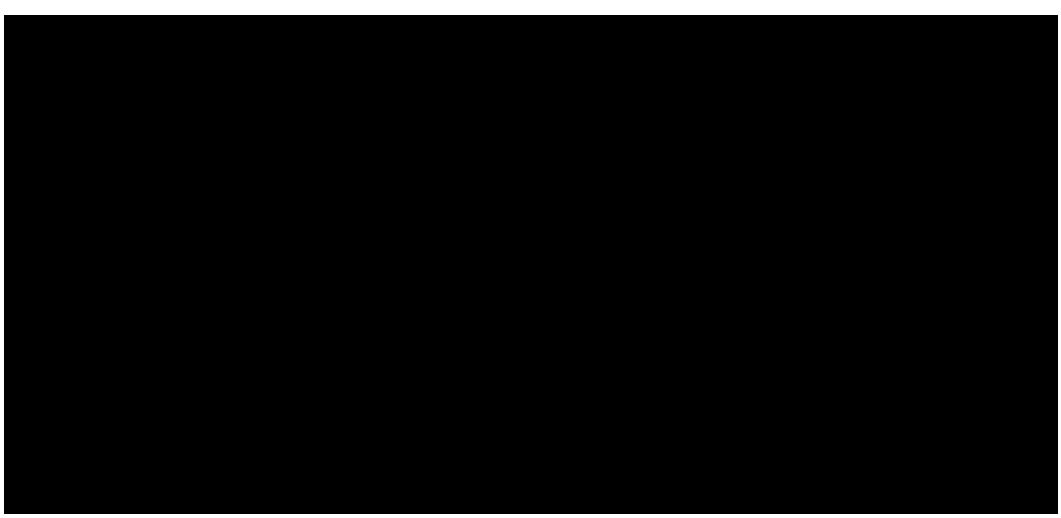

**Figure 5.** Result analysis of the factors driving the growth in fertilizer application from 2000 to 2017.

From 2000 to 2008, the total increase in chemical fertilizer application reached 11.15 million tons. The increase in chemical fertilizer application intensity was the main factor leading to the rapid growth in total chemical fertilizer application, with an average contribution rate of 113.13%. The adjustment of planting structure and the change in sowing area restrained the increase in chemical fertilizer application to a certain extent. Excessive application of chemical fertilizer caused serious environmental pollution in China, and its contribution to water pollution was more than 60.00% [33]. In 2007, the main water pollutant emissions from agricultural sources (excluding rural living sources in typical areas) in China included 13.24 million tons of chemical oxygen demand (COD), 2,704,600.00 tons of total

nitrogen (TN), 284,700.00 tons of total phosphorus (TP), 2452.09 tons of copper, and 4862.58 tons of zinc. Among them, the loss of TN and TP in the planting industry was 1,597,800.00 and 108,700.00 tons, respectively [34].

From 2009 to 2014, there was a slowing down in the growth rate of total chemical fertilizer application, with a total increase of 7.51 million tons in six years. The increase in the planting area of crops became the main reason for the growth in total chemical fertilizer application, with an average contribution rate of 51.88%, slightly higher than that of the fertilizer application intensity at 51.72%. During this period, the sown area of crops increased by 10.94 million ha, with an average annual growth rate of 1.15%, far exceeding the average growth rate of the sown area (0.30%), which was the fastest growth period of sown area of crops in the statistical period (Figure 6).

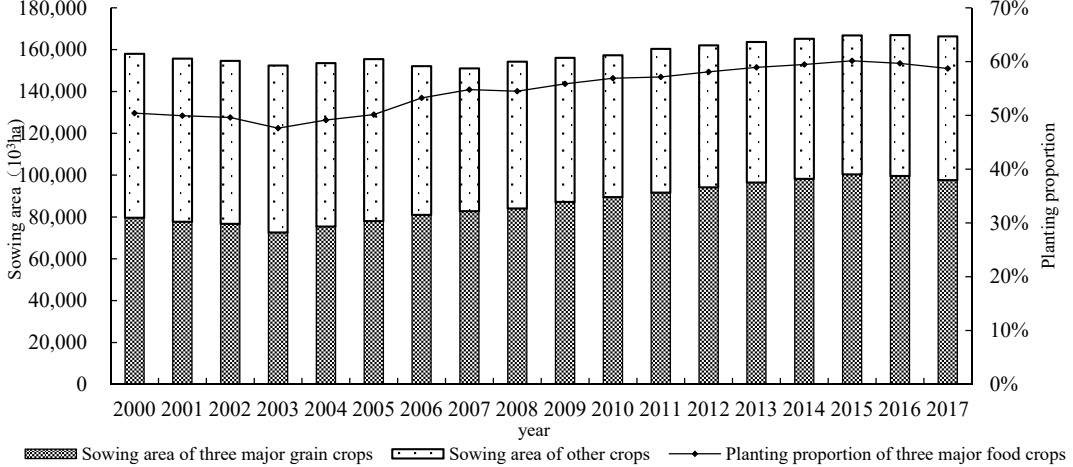

**Figure 6.** Changes in the trend of the planting area and planting structure of crops in China from 2000 to 2017.

In 2015, to reduce costs and environmental pollution caused by the overuse of chemical fertilizer, China began to control the increase in chemical fertilizer application by promoting precise fertilization, adjusting the use structure of chemical fertilizers, improving fertilization methods, amongst other measures [35]. In addition, the Ministry of Agriculture and Rural Affairs of the People's Republic of China and other relevant departments jointly issued the pilot program to explore the implementation of a farmland rotation and fallow system in 2016 to reduce fertilizer application, in which the fallow area would be gradually expanded along with a reduction in the sown area [1]. During 2015–2017, the application of chemical fertilizers showed negative growth. The decrease in the application intensity of chemical fertilizers was the main driving factor, with an average contribution rate of 126.81%. The average contribution rate of the decrease in the sowing area to the decrease in the use of chemical fertilizers was 31.99%.

During the statistical period, the effect of planting structure adjustment on the change in fertilizer application was the lowest mainly because the change in the crop planting structure in China has not been significant in recent years (Figure 6). The major crops accounted for the majority of the consumed fertilizer, occupying more than 45.00% of the sown area.

### 3.2.2. Contribution Rate of the Major Grain Crops to the Growth of Total Chemical Fertilizer Application

The fertilizer consumption of rice, wheat, and corn accounts for more than 50.00% of China's total fertilizer consumption. Thus, the main factor affecting China's total fertilizer demand is still the fertilizer demand of the main grain crops. Based on the decomposition of factors such as application intensity, planting structure, and sown area, we further studied the contribution rates of the major crops to the total chemical fertilizer application. The decomposition results for each time period are shown

in Table 4. Since 2000, the amount of chemical fertilizer application for the major crops cumulatively increased by 10.79 million tons, and the cumulative contribution to China's total chemical fertilizer application reached 62.14%. This was the main reason for the increase in chemical fertilizer application in China, in which the effect of application intensity was the most prominent.

**Table 4.** Analysis of the factors driving for growth in fertilizer application in China's major grain crops from 2000 to 2017.

| Time Period | Total Variation of Fertilizer Application ($10^4$ t) | Total Contribution Rate (%) | Application Intensity | | Planting Structure | | Sown Area | |
|---|---|---|---|---|---|---|---|---|
| | | | Contribution Value ($10^4$ t) | Contribution Rate (%) | Contribution Value ($10^4$ t) | Contribution Rate (%) | Contribution Value ($10^4$ t) | Contribution Rate (%) |
| 2000–2002 | −172.92 | −82.22 | 107.27 | 51.01 | −253.59 | −120.58 | −26.6 | −12.65 |
| 2003–2005 | −34.34 | −8.11 | −65.82 | −15.54 | 17.92 | 4.23 | 13.56 | 3.20 |
| 2006–2008 | 301.31 | 62.55 | 116.25 | 24.13 | 204.95 | 42.55 | −19.89 | −4.13 |
| 2009–2011 | 478.02 | 103.61 | 226.87 | 49.17 | 133.43 | 28.92 | 117.73 | 25.52 |
| 2012–2014 | 381.19 | 131.43 | 149.81 | 51.65 | 131.15 | 45.22 | 100.23 | 34.56 |
| 2015–2017 | 125.33 | −95.53 | 148.77 | −113.40 | −48.29 | 36.81 | 24.85 | −18.94 |
| Cumulative total | 1078.59 | 62.14 | 683.14 | 39.36 | 185.56 | 10.69 | 209.89 | 12.09 |

During 2000–2005, the change in fertilizer application to the major grain crops was the opposite to that of the total change in fertilizer application, showing negative growth, with a cumulative reduction of 2.07 million tons or about one-quarter of the amount of applied fertilizer. This result was mainly due to the adjustment in the planting structure of grain crops.

From 2006 to 2017, the fertilizer application rates of the major crops experienced two periods: first, a period of rapid growth followed by a period of slow growth. This became the main reason for the increase in the total chemical fertilizer application rate, with a cumulative contribution rate of 116.69%. The increase in fertilizer application caused by the increase in fertilizer application intensity of the major crops accounted for 58.25% of the total increase in fertilizer application (Figure 7).

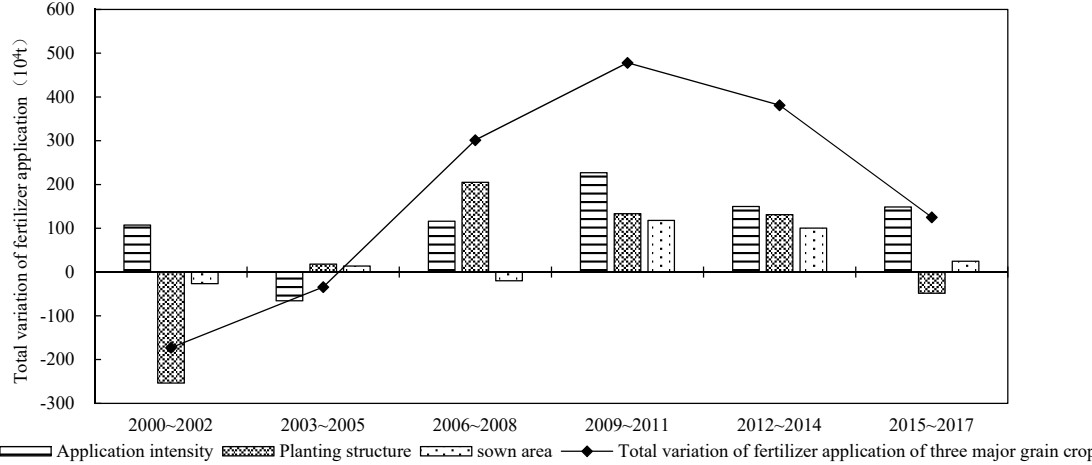

**Figure 7.** Analysis of the factors driving the growth of fertilizer application in China's major grain crops from 2000 to 2017.

*3.3. Environmental Effect of Excessive Use of Chemical Fertilizers and the Target of Chemical Fertilizer Reduction*

The excessive application of chemical fertilizers causes many environmental problems. Chemical fertilizers decompose and volatize easily, and excessive $CH_4$ and $N_2O$ enter the atmosphere, causing photochemical smog and the greenhouse effect [36,37]. Unabsorbed elements in chemical fertilizers (especially nitrogen) enter the surface water through surface runoff, causing water quality deterioration and seriously affecting the survival of water organisms. Nitrogen fertilizers that leach into groundwater cause groundwater nitrate pollution [38,39]. About 60.00% of the nitrogen dissolved in surface water is sourced from chemical fertilizers [4], and the loss of nitrogen and phosphorus in farmland has

become the main factor causing water pollution in China. Excessive or unreasonable application of chemical fertilizers affects the physical and chemical properties of soil and the formation of soil aggregate structure, resulting in soil hardening. The area of farmland polluted by fertilization in China is increasing, and there is nutrient imbalance in most of the available farmland [30,39]. In 2010, the national farmland nitrogen surplus reached 22.40 million tons [40]. Moreover, chemical fertilizers also contain several types of metals, radioactive substances, and other harmful components. Their long-term accumulation leads to indirect pollution of farmland crops [41,42]. About 25.00 million ha of farmland in China is polluted by heavy metals at present, and every year, 12.00 million tons of grain are polluted by heavy metals [42]. The contribution of fertilizer use to farmland pollution and total pollution has increased from 40.00% in the 1980s to 66.00% in 2010, and the direct economic loss accounts for 0.50–1.00% of the national GDP [40].

In accordance with the unified deployment of the Ministry of Agriculture and Rural Affairs of the People's Republic of China, China established an action plan in 2015 to attain zero growth in fertilizer application as an important measure to promote the green development of agriculture, focusing on improving the efficiency of chemical fertilizer use and achieving decoupling to control the use of chemical fertilizers. According to the data published by the Ministry of Agriculture and Rural Affairs of People's Republic of China, the utilization rate of chemical fertilizers for the three major grain crops of rice, corn, and wheat in 2017 was 37.80–2.20% higher than in 2015. However, the amount and intensity of fertilizer use in China remains the highest in the world. In 2017, the amount of fertilizer application per unit of cultivated area in China reached 434.41 kg/ha [26], about three times the average world level, leaving large room for reduction. Starting from the two main sources of growth in fertilizer use, namely application intensity and sowing area, we calculated the reduction target for agricultural fertilizer application in China in which the internationally recognized upper limit benchmark for fertilizer application safety (225.00 kg/ha) could be met, as well as the fallow target (that the fallow area in the rotation area should be more than 3.33 million ha by 2020), formulated by the Ministry of Agriculture and Rural Affairs of People's Republic of China in 2018:

$$RT = Q - SQ \times (A - TA) \tag{8}$$

where RT is the reduction target of chemical fertilizers, SQ is the upper limit of safety of chemical fertilizer application, and TA is the target area of rotation and fallow.

To reach the goal of rotation and fallow in 2020, China needs to reduce fertilizer application by at least 21.80 million tons, a decrease of as much as 37.00%, in order to control the deleterious effects caused by the excessive use of chemical fertilizers on the soil and ecological environments.

## 4. Conclusions and Policy Implications

To solve the problem of environmental pollution caused by the application of chemical fertilizers in China, we used the LMDI method to analyze the trends and factors that have driven the amount and intensity of chemical fertilizer application since the 21st century, and to explore the reason for the growth in chemical fertilizer application in China.

From 2000 to 2014, the amount and intensity of chemical fertilizer application in China continued to increase, but the effect on increasing the grain crops' yield was limited. In 2015, by promoting precise fertilization, adjusting the use structure of chemical fertilizers, and improving the method of fertilization, the amount and intensity of chemical fertilizer application fell for the first time, but the application intensity was still much higher than the world average. The PFP of the main grain crops increased by 1.84 kg/kg in the studied 18-year period, but a large gap remained compared with in developed countries. In terms of driving factors, from 2000 to 2008, the increase in the intensity of chemical fertilizer application was the main factor that led to the rapid growth in the total amount of chemical fertilizer application. The adjustment of the planting structure and the change in the sown area inhibited the growth of chemical fertilizer application to a certain extent. From 2009 to

2014, the increase in crop planting area became the main reason for the increase in the total amount of chemical fertilizer application, which was slightly higher than the average contribution rate of chemical fertilizer application intensity. From 2015 to 2017, the reduction in chemical fertilizer application intensity once again became the main factor driving the reduction in chemical fertilizer application. The amount of chemical fertilizer applied to the three major grain crops accounted for the majority of the increase in the total amount of chemical fertilizer application, with a cumulative contribution rate of 62.14%, with application intensity being the most prominent factor. To achieve the targets for rotation and fallow in 2020, China needs to reduce chemical fertilizer application by at least 21.80 million tons, a decrease of as much as 37.00%.

Based on this research, we provide the following two suggestions to reduce chemical fertilizer application toward protect the environment:

(1)　The key to reducing chemical fertilizer use in China is to reduce the application intensity of chemical fertilizers, especially for the main grain crops. The government should improve the fertilizer management system in accordance with laws and regulations relevant to the fertilizer application system, aiming at overall environmental protection and agricultural energy conservation, actively promoting and applying technology to reduce fertilizer application and increase its efficiency, with major crops being at the core of these efforts. Soil testing and formula fertilization, using livestock and poultry manure as fertilizer, and scientifically based fertilization approaches should also be promoted to improve the use efficiency of chemical fertilizers, reduce the application intensity of chemical fertilizers, and accelerate the decoupling of grain production and chemical fertilizer application.

(2)　The reduction of crop planting area and the adjustment of planting structure can be effective in controlling chemical fertilizer application, which has potential to reduce the application of chemical fertilizers. China should promote the structural reform of the agricultural supply side, attaching importance to implementation of the fallow rotation system of cultivated land, popularizing the planting structure to reduce fertilizer use and improve the contribution of the planting area and structure of agricultural crops in lieu of increasing the amount of fertilizer application.

**Author Contributions:** Conceptualization, R.H.; Formal analysis, R.H.; Funding acquisition, C.S. and R.S.; Investigation, R.H., Z.Z., and R.Z.; Methodology, R.H.; Project administration, C.S. and R.S.; Resources, C.S. and R.S.; Supervision, C.S.; Validation, Z.Z. and R.Z.; Writing—original draft, R.H. and Z.Z.; Writing—review & editing, C.S. All authors have read and agreed to the published version of the manuscript.

**Funding:** This work was supported by the National Key Research & Development Program of China under Contract (No. 2016YFD0201200).

**Conflicts of Interest:** The authors declare no conflict of interest.

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
