# Peer review of "Development Trend and Driving Factors of Agricultural Chemical Fertilizer Efficiency in China"

_sustainability, doi:10.3390/su12114607_

Round 1

Reviewer 1 Report

The article formulates the problem clearly and suggests solutions in the discussion section.
The use of fertilizers and their impact on the environment and systemic approaches in Chinese agriculture are revealed in the article.
The article does not offer new and unknown ideas, but clearly and systematically demonstrates the depth of the problem, possible solutions.

Minor shortcomings:
In some places china is written in lower case (lines 20; 22; 26);
Review the use of English;
Provide more analytical material in article - analysis of the scientific literature, the latest articles on the subject.
In addition, there is a lack of an additional or extended figure 2 which can also include other countries for comparison (USA, Japan, the Netherlands, Germany, France). it would be good if you presented it in article and provide comparison of the use of fertilizers from other countries in relation to the yield obtained.

Reviewer 2 Report

  • The manuscript didn't consult and present sufficient scientific arguments. 
  • I suggest the manuscript should be edited with professional or natives English editors. 
  • Results and Discussion section not provided.
  • several inconsistencies? see the detailed comments. I feel not cached all.  

Specific comments:

Title and affiliation

  • L2: I feel the phrase “changing characteristics…” is misleading. I suggest replacing with other words like Trend or other suitable words
  • L6, L8 and L10: Spacing required after “…China; …” in each line.
  •  

Abstract:

  • The aim of the study was not clearly stated in the abstract.
  • L21: “…growth of the total amount…”
  • L20, L22, L26: “…china..”, change to C
  • L27-31: rewrite or improve it. For example, “On the basis of….”. Can be “Based on the….”. Also, why some suggestions? Remove “some suggestions”. Mention them as only a few intervention options are available.

Introduction

  • L37: what is the source of this evidence? Please provide the ref. or if ref 1&2 are the source, cite properly.
  • L43: Fix misplaced quotation or apostrophe symbol. “…Nations ‘.”
  • L46: Ref 6 is not from EU. Please make sure to properly cite the references.
  • L50: “…and other national conditions.” What are the other conditions? Political, economic ….. it has to be clear.
  • L54: I do not understand what this means “…variation characteristics and influencing factors…”
  • L65-69. Must be rewritten or breakdown into sentences per the idea. Very hard to understand.

Methods

  • Number PEP formula as eq.1.
  • Fix punctuation errors like coma, period (see L119), spacing.. remove unnecessary comas after equation 1-6.
  • L128-129: What is the source of this information (ref)?
  • Cite the sources od databases like IFA, World Bank
  • What is IFA stand for? It has to be clear for the reader.
  • L142: “Despite some errors in the analysis,…” what do you mean? Provide the sources of errors if any? In the same line “.., it can…”. refers to what?
  • May be good to introduce abrevati for “Ministry of Agriculture and Rural Affairs of People’s Republic of China” or consider to use “Ministry of Agriculture and Rural Affairs, MARA”.
  • Please provide more justification and references why you choose only 3 crops or 31 provinces/cities.
  • I suggest the methods section must be revised for better understanding.

Results:

  • Please add result and discussion section
  • L171: “After the original Ministry…” what do you mean by “original”?
  • L180 “…39%, 32.62%, 18.16% and 22.79%, respectively.” Where did you bring this numbers? If calculated from table 1, add row at the bottom of the table to show the fertilizer use change. Whenever you provide numbers/stat you must refer to the results (table or figures).
  • Figure 1: kg/ha is the standard SI and most common one, please consider using kg/ha. The fig 1 title lacks a “period” at the end.
  • In table 1 “USA”; in fig 1 key “U.S.A”. please be consistent.
  • L196: “…year by year..” do you mean annually? Considering using “…the grain output was annually declined…”
  • Most of the discussions inclined towards opinions, it is good to support with evidence and references. For example, L201-203 and many others. Please consider strengthening your arguments.
  • L206-210 is repetition of L201-205? Please recheck and avoid redundancies.
  • L211-215 is not clear. Must be rephrased. Several sentences in the manuscript like this one are 4-5 lines long. Please try to restructure your sentences so the readers easily grasp the messages.
  • L228: correct the significant digits “…58.594 million tons.” to 58.59 million tons. Be consistent with sig digits. Please fix other numbers in the manuscripts by depending on one style. Are the “tons” in this manuscript referring metric unit (1t=1000 kg)? If so, tonne is the proper unit.
  • L261: be consistent when using thousand tons or kilotons?
  • L276: better to use ha instead of “thousand ha”.
  • L278: remove ‘see’ from “…period (see Figure 5).”
  • L294, L296, L298 and L318: Table 2, Table 3 and Figure 2 title requires ‘period’ at the end of the title. I am not sure if the “accumulative total” is the correct phrase in Table 2, consider replacing with ‘cumulative total’.
  • I think it is unnecessary to use “, …, as shown in Figure 6.” replace with (Figure 6).
  • Be consistent with using “fertilizer” or “fertiliser”. See fig 6 y-axis title and fig title (L321).
  • L325-327 must be rephrased. correct the subscript of the methane and nitrous oxide chemical formula.
  • L327: Inappropriate phrase “Unused chemical fertilisers….” What do you mean? Please rephrase it, it is not fertilizer entering the water bodies. Please consult literatures.
  • L328: what do you mean “…other ways…”? is it leaching, …
  • L330: unnecessary phrase “According to statistics,..”. Just start with “About 60% …”
  • L331: which water? Not clear
  • L335: “are out of balance” what do you mean?
  • L338: “it causes serious farmland soil pollution”, ambiguous phrase?
  • Consider rephrasing L339.
  • L341: I don’t think using “last century” is appropriate? Better use a specific timeline like during1960s
  • L348-349: “…three major grain crops of rice, corn, wheat….” After introducing the three major crops. I think “…major crops in 2017…” is enough. As we know which are the major crops. “three major grain crops” was unnecessarily repeated in few places. Consider rephrasing the sentence.
  • L356: “…50 million mu by 2020” is the mu is SI unit.
  • Number this equation as “RT=Q-SQ×(A-TA)”…(8)
  • L363 and L395: Not proper to say “..long way to go.” Please rephrase these sentences.

Conclusion:

  • L400-408: is very long sentence. Consider breaking into 2-3 sentences.
  • Also, L413-420 is a very long, fragmented sentence, hard to understand. I suggest to change “…our country, we…” with the country or China. Also, this sentence looks personalized opinion. I suggest rephrasing the sentence.

Funding:

  • What R & D stands for, make it clear for the reader?

References:

  • L432 & L521: “..c.” and also spacing after a period is required after L432.
  • L442: spacing required before “Effect…”
  • L448: Correct “…waterpollution..” to water pollution.
  • L483: spacing required before “nutrient…”
  • L498: spacing before “state…”
  • L493 and L505, “In Chinese” and “in Chinese”. Be consistent. I suggest using (Chinese).
  • L520: spacing before “existing …” and what id “…farml…”

Round 2

Reviewer 2 Report

Thank you for addressing my comments and suggestions. However, I am not satisfied with the language. Please the manuscript has to be language proofed as there are some grammatical and punctuation errors. Also, inconsistent significant digits and capitalization. Some of the sentences are still long and hard to read.   

  For example: 

L20: ...that (1) A large gap...

L30: ...least 21,800 thousand tonnes... please correct all inconsistent use of significant digits across the manuscript.  

L53-56: ...zero growth of chemical fertilizer use.. this repeated in the same sentence. can be rearticulated to avoid redundancy of ideas. 

L136: ....(database(IFA)[21], the World Bank’database(TWB)[22]... spacing issue.

L

L139: ....China Statistical Yearbook(CSY, 2001-2018)[26](table 1). Capitalization.

L142: .....data (2017), and....China (2015). The years in the bracket are confusing and don't indicate the year of the data.

L171: ....between 110 and 140 kg/ha,... .....was 503.32 kg/ha.... inconsistent decimals.

L182-183: ..the increase of main grain yield, but the its promotion effect in China was relatively poor.

L189: ....5627.08kg/ha,... spacing

L244: ....than 24 and 18 kg/kg, and the PEP of corn also remained around 30 kg/kg.

L245: "There is a large gap exists between China’s PEP and that of developed countries [32]." what are the possible reasons? or the comparison of developing and developed countries might be wrong?

L294: ....more than 45% of the....

L317: It becamed the main reason for the increase...

L352: ....2017 was 37.8%, 2.2 percentage points higher...

L376: ....grain  crops increased by 1.84 kg/kg.... 

L408: 5. Patents?? is this a patented work or why? or you want to say contribution?

L418: "References" need to number as 6. References 

Round 3

Reviewer 2 Report

Dear Authors,

Thank you for taking the comments seriously and fixing most of the typo, language, and structure issues. 

Sincerely 

Author Response

     We are very grateful to you for taking the time to carefully review our manuscript and every revision!

     According to the your comments,we have revised the whole paper repeatedly, and our manuscript have been  also submitted to MDPI for English editing. I would like to thank you again for your valuable time and energy in reviewing, making comments and reviewing the revised manuscripts, which have helped us to improve our article and make it better. Next, we will continue our existing research more seriously and enjoyably, hoping to contribute a little to the improvement of the environment.Thank you.